# TALEN-Mediated Gene Targeting for Cystic Fibrosis-Gene Therapy

**DOI:** 10.3390/genes10010039

**Published:** 2019-01-11

**Authors:** Emily Xia, Yiqian Zhang, Huibi Cao, Jun Li, Rongqi Duan, Jim Hu

**Affiliations:** 1Translational Medicine Program, Hospital for Sick Children Research Institute, 686 Bay Street, Toronto, ON M5G 0A4, Canada; emily.xia@mail.utoronto.ca (E.X.); yiqian.zhang@mail.utoronto.ca (Y.Z.); huibi.cao@sickkids.ca (H.C.); jun.li@sickkids.ca (J.L.); cathleen.duan@sickkids.ca (R.D.); 2Department of Laboratory Medicine and Pathobiology, University of Toronto, 1 King’s College Circle, Toronto, ON M5S 1A8, Canada; 3Department of Paediatrics, University of Toronto, 1 King’s College Circle, Toronto, ON M5S 1A8, Canada

**Keywords:** cystic fibrosis, gene therapy, site-specific gene targeting, viral vector, TALEN

## Abstract

Cystic fibrosis (CF) is an inherited monogenic disorder, amenable to gene-based therapies. Because CF lung disease is currently the major cause of mortality and morbidity, and the lung airway is readily accessible to gene delivery, the major CF gene therapy effort at present is directed to the lung. Although airway epithelial cells are renewed slowly, permanent gene correction through gene editing or targeting in airway stem cells is needed to perpetuate the therapeutic effect. Transcription activator-like effector nuclease (TALEN) has been utilized widely for a variety of gene editing applications. The stringent requirement for nuclease binding target sites allows for gene editing with precision. In this study, we engineered helper-dependent adenoviral (HD-Ad) vectors to deliver a pair of TALENs together with donor DNA targeting the human AAVS1 locus. With homology arms of 4 kb in length, we demonstrated precise insertion of either a *LacZ* reporter gene or a human cystic fibrosis transmembrane conductance regulator (*CFTR*) minigene (cDNA) into the target site. Using the LacZ reporter, we determined the efficiency of gene integration to be about 5%. In the CFTR vector transduced cells, we were able to detect CFTR mRNA expression using qPCR and function correction using fluorometric image plate reader (FLIPR) and iodide efflux assays. Taken together, these findings suggest a new direction for future in vitro and in vivo studies in CF gene editing.

## 1. Introduction

Cystic fibrosis (CF) is an inherited autosomal recessive disease most commonly seen in the Caucasian population [1]. Cystic Fibrosis is caused by mutations in the cystic fibrosis transmembrane conductance regulator (CFTR) gene. The most predominant mutation leads to the deletion of the phenylalanine residue at position 508 [2]. Cystic Fibrosis affects multiple organs and currently 80% of mortality is caused by lung failure [3]. Conventional treatments including antibiotics, physical therapy, and nutritional supplements can alleviate the symptoms but not provide an effective treatment for the disease. Over the past few years, new treatment strategies have been evolving; different types of CFTR channel modulators have been shown to be effective in improving channel activity in CF patients [4,5,6,7,8,9,10]. Most recently, a triple combinational therapy with VX-440, tezacaftor, and ivacaftor has shown significant clinical benefits in an ongoing phase II trial [11]. While substantial progress has been made in the development of CF therapies, current treatment strategies including the multiple channel modulator/corrector usage impose a heavy drug burden on patients. In addition, for a small portion of patients whose mutations lead to no CFTR protein production, there is no effective drug to slow down the disease progression. Furthermore, possibilities of drug interactions may increase the risk of adverse effects [12].

Gene therapy is an attractive strategy for CF lung disease because it treats the underlying cause of the disease rather than its symptoms. Gene replacement therapy for mutations involved in Leber’s congenital amaurosis, choroideremia, achromatopsia, and retinitis pigmentosa using adeno-associated viral AAV vectors has shown correction to various degrees [13,14,15]. Many CF gene therapy trials using adenoviral (Ad) vector, AAV vector, liposomes as vehicles for delivery have been carried out since the early 1990s, but none has shown significant improvements in lung function [16,17,18,19,20]. Nonviral vectors, such as liposomes, are not efficient in gene delivery in general while viral vectors have other problems [21]. Recent studies show improvement in non-viral vector delivery [22,23], but the methods have yet to be tested in large animal models. AAV vectors have been successfully used in eye gene therapy [14], but it has a small DNA-carrying capacity and is thus not suitable for delivering a CFTR gene expression cassette. For Ad vectors, one major obstacle is the host immune responses elicited by the vectors, which eliminates the vector-transduced cells [24]. To reduce the host immune responses, a helper-dependent adenoviral (HD-Ad) vector has been developed by deleting all viral coding sequences. Thus, this type of vector has a large DNA-carrying capacity, up to 36 kb, in addition to advantages in reduction of the host immune responses [25,26]. Using HD-Ad vectors, our group has demonstrated efficient vector delivery to airway epithelia of mice and pigs with limited immune responses [26,27,28]. More recently, we have shown effective delivery of HD-Ad vectors to airway basal cells of mice and pigs [29].

Engineered transcription activator-like effector nucleases (TALENs) are highly efficient in generating double-stranded breaks in cell genomes, which facilitate gene editing via the homology-directed recombination (HDR) process [30]. TALENs bind and cleave their target DNA as heterodimers. Each TALEN monomer is composed of a FokI nuclease fused to a customizable DNA-binding domain [30]. TALEN has been applied in gene editing both in vitro and in vivo [31,32,33]. Notably, Suzuki et al. have shown TALEN-mediated editing for F508del in induced pluripotent stem cells (iPSCs) [34]. Unlike the more popular CRISPR-cas9 system that recognizes a 20 bp target [35], TALEN recognition sequences can be designed longer to provide better specificity and they are commonly designed to be 34 bp in length [36,37]. 

We have engineered a pair of TALENs that target the human adeno-associated virus integration site 1 (AAVS1) with assistance from Dr. Bo Zhang group at Peking University, China. Because of the large carrying capacity of the HD-Ad vector, we have packaged both the TALEN and donor into one single vector. This all-in-one vector delivers both TALENs and donor DNA together to target cells with a high efficiency. We have demonstrated that a LacZ reporter donor gene expression cassette can be successfully integrated into the AAVS1 locus. Furthermore, we have also shown that permanent integration of a *CFTR* minigene [38] can be achieved in cultured airway epithelial cells. These findings collectively demonstrate feasibility of using TALENs for future targeted *CFTR* gene editing in vivo.

## 2. Result

### 2.1. Co-Delivery of TALEN and Donor DNA Using HD-Ad Vectors

The human AAVS1 locus has been extensively explored for targeted gene integration studies as it leads to abundant transgene expression with no adverse consequences [39,40]. To test if large gene constructs, such as a *CFTR* expression cassette, can be integrated into this locus, we first tested integration of a LacZ reporter gene expression cassette (Figure 1). For homology-directed gene targeting, homology arms of 4 kb in length were added to either end of the LacZ gene expression cassette (Figure 1A, Appendix A). The LacZ HD-Ad vector that harbors both the donor and TALEN genes as well as the GFP gene between two TALEN homodimers (Appendix A) was produced as described [41]. IB3-1 cells were transduced at passage 0 and GFP served as a reporter for transduction efficiency (Figure 1B). The TALEN cleavage efficiency was evaluated using the T7E1 assay [42] (Figure 1C). It was around 36% when cells transduced at 50 MOI (multiplicity of infection or infectious vector particles per cells) or 66% when cells transduced at 100 MOI.

### 2.2. Analysis of the LacZ Reporter Gene Integration into the AAVS1 Locus

Junction PCR assays indicated precise on-target integration of the LacZ reporter gene into the AAVS1 locus (Figure 2A). Junction PCR products were further subjected to analyses using restriction enzyme (RE) digestion and Sanger sequencing (Figure 2B–D), both of which further confirmed the identity of junctional PCR products.

To determine the integration efficiency of LacZ vector, transduced cells were first continuously passaged for 18 generations (Appendix A). Passaging was implemented to dilute out residual vector genomes in transduced cells. The LacZ gene integration efficiency was determined via staining for the percentage of LacZ-positive cells at passage 18 (Figure 3A,B). The addition of SCR7 (a DNA ligase inhibitor) [43] has been shown to enhance gene integration by promoting cellular repair via the homology directed repair (HDR) pathway [44]. In line with this finding, we have also observed an increase of 1.5-fold in the integration efficiency with SCR7 treatment (Figure 3C). To verify the gene integration efficiency with an independent method, Imagene Green substrate was used to fluorescently label LacZ positive cells at passage 18, and the percentage of LacZ positive cells was quantified by flow cytometry (Figure 3D). Integration efficiencies determined from LacZ staining were 3% for 50 MOI transduction, and 5% for 100 MOI transduction; meanwhile, the flow cytometry showed similar integration efficiencies of 3.6% for 50 MOI transduction and 5.5% for 100 MOI transduction.

### 2.3. Integration of CFTR Gene into the AAVS1 Locus Allowed Persistent CFTR Expression and Presence of CFTR Channel Activity

For the *CFTR* gene integration, an HD-Ad vector that contained *CFTR* cDNA expression cassette [38] flanked by homology arms in addition to TALEN genes was constructed and produced (Figure 4A, Appendix A). The enhanced green fluorescent protein (EGFP) gene was inserted between two TALEN homodimers, which served as a reporter for successful transduction (Appendix A). Following transduction, the CFTR protein was expressed in transduced IB3-1 cells (Figure 4B, Appendix A). Junction PCR on either end of the integration site showed positive products, indicating the integration was at the precise location (Figure 4C). The PCR product was validated using restriction enzyme digestion and Sanger sequencing (Figure 4D–F). We further examined *CFTR* mRNA expression at multiple time points after transduction by qPCR, and found that the *CFTR* mRNA level was significantly higher in cells transduced with the integration vector than that in cells transduced with the control (non-integrating) vector (Figure 5A). More importantly, CFTR protein was detectable at passage 6 and passage 12 for cells transduced with CFTR integration vector; whereas cells transduced with non-integrating vector failed to show detectable *CFTR* protein expression at passage 12 (Figure 5B).

To assess the level of CFTR channel activity, we performed fluorometric image plate reader (FLIPR) assays in 96-well plates with cells treated with *CFTR* integration vector. A FLIPR assay was used to monitor the apical membrane chloride conductance mediated by CFTR channel [45]. Results indicated significant CFTR channel activity, both at initial transduction and at passage 12 (Figure 5C). This result was consistent with CFTR channel activity measured using iodide efflux assay [26] (Figure 5D). Looking at these findings, it is evident that our vector could achieve effective integration of the *CFTR* expression cassette into the AAVS1 locus and allow persistent CFTR expression at mRNA, protein, and functional level.

### 2.4. Transient TALEN Expression

Because TALEN is a foreign protein and is not naturally expressed in humans, its prolonged expression in hosts may cause unwanted antigenic responses. We therefore examined TALEN expression over time in cells transduced with a *CFTR* integration vector. We first monitored the GFP fluorescence level following vector transduction for 12 days and noticed a significant decline in GFP fluoresce after day 3 (Figure 6A). We then performed qPCR and Western blotting to measure TALEN expression in transduced cells at multiple time points following integration vector transduction. Results showed that TALEN protein and mRNA expression was undetectable after 5–6 passages (Figure 6B,C, Appendix A). In addition, the relative quantity of HD-Ad vector genome measured using qPCR also gradually declined and was undetected after six passages (Figure 6B, Appendix A). These results collectively indicate that although high levels of TALEN expression can be achieved upon initial administration of the gene integration vector in cells, its expression was lost gradually in transduced cells. Hence, we conclude that the HD-Ad integration vector did not show prolonged presence of TALEN or vector in transduced cells.

## 3. Discussion

In this study we showed insertion of large 8 kb gene constructs into the human AAVS1 locus can be achieved at an efficiency of 5%. Because of the large carrying capacity and high transduction efficiency, both TALEN and donor genes can be simultaneously delivered by a single vector. This method of co-delivery of donor and gene editor cannot be achieved by other viral vectors due to their limited carrying capacity. In addition, our laboratory has also demonstrated that HD-Ad vectors can target the residential progenitor cells in the airway system [29]. This will allow us to explore the possibility of *CFTR* gene editing in vivo in progenitor cells for sustained *CFTR* gene expression in the airway system in our future work.

In gene editing studies, one of the crucial safety criteria is the targeting precision. Off-target cleavage at essential genes may cause deleterious consequences. One limitation of this study is that we have not investigated the off-target effects of our gene correction strategy. Studies have revealed that small point mutations such as substitutions and deletions caused by DSB may allow cell transformation into an oncogenic nature [46,47]. The challenge is faced by all the endonuclease-mediated gene editing systems, including, zinc fingers, TALENs, and CRISPRs [48]. The CRISPR-cas9 system has been used widely as a versatile tool for gene editing [49]. Compared to the CRISPR-cas9 system, which shows off-target effects due to the tolerance for mismatched binding of its short 20 nt recognition sequence [49], TALEN recognition requires a longer binding sequence, which minimizes the chance of off-target cleavage [50]. Genome wide off-target studies looking for cleavage at sequences similar to on-target site revealed lower off-target cleavage for TALEN than for the CRISPR-cas9 system [51,52,53,54]. Technical advancements in the field will likely improve the safety profile of all these gene editing systems.

We chose to integrate the *CFTR* expression cassette into the AAVS1 locus, instead of editing a *CFTR* mutation as our strategy for *CFTR* gene correction because this approach is not mutation-specific and it is applicable to correcting all *CFTR* mutations. We selected the AAVS1 locus for the test because it is known that transgenes integrated in this locus can be efficiently expressed. However, future work should consider targeting the *CFTR* locus with this approach since integration of a CFTR expression cassette in the locus can stop the mutant CFTR gene expression. This may be considered an advantage over gene integration into a non-CFTR locus.

The efficiency of site-specific gene integration is important for achieving therapeutic effects in gene therapy. SCR7 treatment has been used previously by other groups to increase CRISPR-cas9 mediated integration efficiency. We tested different concentrations of SCR7 for its effect on efficiency of gene integration. We did see an increase in integration efficiency on TALEN mediated gene editing although the effect was not as significant as mentioned previously by other groups [44,55]. One reason could be the difference in gene editors utilized as other groups mainly applied SCR7 in combination with the CRISPR-cas9 system. Another reason could be the difference in the cell lines used. We are currently working on enhancing the efficiency of *CFTR* gene integration. We also want to explore the possibility of combining protein factors [56] that facilitate the HDR pathway with TALEN integration vectors.

In this study, we examined the time course of TALEN expression in transduced cells. Since TALEN was derived from a pathogenic bacterial species from plant Xanthomonas (genus) [36], we worry that it may cause antigenic responses in the host, which are a significant problem seen in studies of in vivo gene therapy [25]. To our favor, this study has shown a rapid clearance of TALEN in the transduced cells, examined at the levels of mRNA and protein. This greatly limits the chance of host immune responses that may cause elimination of gene-corrected cells. Taking all of these together, our study demonstrated successful TALEN-mediated gene correction in a human *CFTR* mutant cell line, as well as rescuing of the *CFTR* channel activity. This provided new insights in TALEN-mediated gene editing and opened opportunity for further enhancing the efficiency and performance in future studies.

## 4. Material and Methods

### 4.1. Cell Culture

IB3-1, a cystic fibrosis cell line was derived from bronchial epithelial cells of a CF patient with one F508del allele and one W1282X allele via immortalization with adenovirus 12 SV40 virus hybrid (Ad12SV40) [57]. IB3-1 was maintained in DMEM medium (Life Technologies Canada, Mississauga, ON, Canada) containing 10% heat-inactivated FBS (Wisent, Saint-Jean-Baptiste, QC, Canada) and 100 U/mL penicillin-streptomycin (Life Technologies Canada, Mississauga, ON, Canada).

### 4.2. Plasmid Constructs

TALEN plasmids were generated by Dr. Bo Zhang’s lab using the Unit Assembly method [31]. The RVDs HD-HD-HD-HD-NG-HD-HD-NI-HD-HD-HD-HD-NI-HD-NI-NN-NG and NG-NG-NG-HD-NG-NN-NG-HD-NI-HD-HD-NI-NI-NG-HD-HD-NG recognize a region in the AAVS1 site within the human chromosome 19 with a 15-nucleotide spacer. Generation of the UBCLacZ expression cassette was as described in the previously [58]. Left and right homology arms (4 kb) were amplified from human cell line A549 (ATCC CCL-185) and cloned into the UBCLacZ vector by in-fusion cloning (Clontech, Mountain View, CA, USA). The UBCLacZ expression cassette with homology arms was cloned into an HD-Ad backbone plasmid pC4HSU (NotI/SalI) [59]. The PmeI sites on the resulting vectors were switched to PacI sites using infusion cloning. The assembled UBCLacZ plasmid pC4HSU-UBCLacZ-L4-R4 was linearized with NheI followed by a blunt end treatment with T4 DNA polymerase (New England Biolabs, Whitby, ON, Canada), then digested with NsiI. The DNA fragment for TALEN expression was cleaved and ligated using FspI and Nsilsites. The resulting vectors were denominated as pHD-Ad-UBCLacZ-TALEN.

The same methods as described above were used to add the 4 kb homology arms to the K18CFTR expression cassette carried by the pBSII-SK(+) backbone plasmid. Similarly, the expression cassette and the homology arms were cloned into the pC4HSU backbone (NotI/SalI). The DNA fragment for TALEN expression was inserted to the pC4HSU-K18CFTR-L4-R4 using AscI digestion and ligation. The resulting plasmid was denominated as pHD-Ad-K18CFTR-TALEN.

### 4.3. Vector Production

HD-Ad-UBCLacZ-TALEN and HD-Ad-K18CFTR-TALEN vectors were produced as previously described [41]. Briefly, HD-Ad vectors were packaged by transfecting 116 producer cells with the linearized pHD-Ad vector and transduced with a NG163 helper virus. In 116 cells, HD-Ad vectors were amplified using a serial passage and then purified using CsCl density gradient ultracentrifugation. Viral particle numbers of HD-Ad preparations were determined using spectrophotometry (Beckman Coulter, Indianapolis, Indiana, USA).

### 4.4. Transduction

IB3-1 cells were seeded in six-well dishes and cultured until 70% confluency. Before transduction, cells were washed with pre-warmed PBS, pH 7.4. HD-Ad vectors were added to the cells at 50 MOI and 100 MOI in 0.5 mL of serum-free media. Cells were incubated for 1 h before adding pre-warmed media to make a final volume of 2 mL per well. Transduced IB3-1 cells were cultured for five days before the first passage. After the first passage, the cells were split twice per week at a ratio of 1:7.

### 4.5. Measurement of Cleavage Efficiency

Cells were collected and lysed using the Genomic Cleavage Detection Kit (Thermo Fisher, Waltham, MA, USA) 72 h post-transduction. A 644 bp region covering the TALEN target site was amplified; 5′ GAT CCT CTC TGG CTC CAT CGT AAG CAA AC 3′ (forward primer) and 5′ GAT GGC CTT CTC CGA CGG ATG TCT C 3′ (reverse primer). The PCR products were denatured at 95 °C for 5 min, followed by re-annealing at a ramp rate of −2 °C/s from 95–85 °C and −0.1 °C/s from 85–25 °C. The T7 enzyme was added and the mixtures were incubated at 37 °C for 30 min before the reactions were terminated by adding 1.5 μL of 0.25 M EDTA. Resulted DNA bands were visualized on a 2% agarose gel. The band intensities were analyzed with ImageJ (NIH, Bethesda, Maryland, USA).

### 4.6. Verification of On-Target Integration

Genomic DNA of cells were collected 5 days post-transduction. Junction PCR was performed on a SimpliAmp Thermal Cycler (Applied Biosystems, Foster City, CA, USA). The sequences of primer sets used and the size of amplicons are as follows: UBCLacZ left junction, 4.5 kb amplicon, 5′ CAT CAG CGA TGC AAT GAT GCT TGG GTT TGC ACC AAT G 3′ (forward primer), 5′ TCC TTC TGC TGA TAC TGG GGT TCT AAG GCC GAG TC 3′ (reverse primer); UBCLacZ right junction, 4.4 kb amplicon, 5′ GGT TTT TCA CAG ACC GCT TTC TAA GG 3′ (forward primer), 5′ GTT GGA GGA GGA AGG AGA CAG AAT CC 3′ (reverse primer). K18CFTR left junction, 4.6 kb amplicon, 5′ CAT CAG CGA TGC AAT GAT GCT TGG GTT TGC ACC AAT G 3′ (forward primer), 5′ GGC AGA GCA CAG ATA AAG AGC CTG AGC CTG GAT TG 3′ (reverse primer); K18CFTR right junction, 4.7 kb amplicon, 5′ GAA TTC GAT GTG CTG GGA TCA GGA G 3′ (forward primer), 5′ GTT GGA GGA GGA AGG AGA CAG AAT CC 3′ (reverse primer). PCR products were purified and digested with EcoRV or AfeI. Sanger sequencing was performed by the Centre for Applied Genomics at the Hospital for Sick Children.

### 4.7. SCR7 Treatment

The NHEJ inhibitor SCR7 (Xcess Biosciences, San Diego, CA, USA) was added at 0.1 and 0.5 μM directly to IB3-1 cells transduced with HD-Ad-UBCLacZ-TALEN 20 h post-transduction. Treated cells were cultured until day five after the transduction and passaged as normal.

### 4.8. β-galactosidase Staining for Integration Efficiency

Transduced IB3-1 cells were washed with PBS, pH 8.0, and fixed in 0.5% glutaraldehyde in PBS for 15 min. After PBS washing, a β-galactosidase staining solution (0.1% X-gal, 2 mM MgCl_2_, 5 mM K-ferricyanide, 5 mM K-ferrocyanide in PBS) was added to cover the cell monolayer. Cells were incubated at 37 °C overnight in the dark. To terminate the reaction, the staining solution was removed from the dish and the cells were washed with PBS.

To monitor the transduction efficiency, IB3-1 cells were stained three days post-transduction. To assess the integration efficiency, IB3-1 cells were passed for 18 passages before staining. Twenty images were taken randomly for each well under a bright field microscope at 100× magnification. The number of LacZ positive cells and the total number of cells in each image were recorded with ImageJ. The transduction and integration efficiencies were calculated according to the following formulas: % transduction = 100 × sum of LacZ positive cells from all 20 images/sum of total number of cells from all 20 images; % integration = 100 × sum of LacZ positive cells from all 20 images/(sum of total number of cells from all 20 images × transduction efficiency).

### 4.9. Single-Cell Colony Analysis

Transduced IB3-1 cells were cultured for 12 passages before single cell sorting. Upon 70% confluency, the cells were suspended in ice-cold PBS, pH 7.4, containing 1% FBS. The cells were incubated on ice in the dark with 7-AAD (BioLegend, San Diego, CA, USA) at a concentration of 100 ng per million cells for 10 min. Single cells were sorted directly into 96-well plates containing 20% DMEM medium supplemented with 20% FBS (MoFloXDP BRV/UV). Sorted cells were cultured for two weeks until assaying.

### 4.10. Flow Cytometry

Upon 90% confluency, transduced IB3-1 cells at passage 18 were gently washed with PBS, pH 7.4, scraped from the plates in 1 mL of PBS, and spun down at 400 rpm for 1 min at room temperature. The cell pellet was resuspended in 300 μL of DMEM, high glucose, HEPES, no phenol red (Life Technologies Canada, Mississauga, ON, Canada) with 3 μL of 30 mM chloroquine diphosphate, and incubated at 37 °C for 30 min. A total of 3.3 μL of 1:10 diluted ImaGene Green C_12_FDG substrate reagent (Invitrogen, Waltham, MA, USA) was added and samples were incubated 37 °C for 45 min in dark. Then, cells were washed and suspended in 500 μL of media containing 500 µM of phenyl-ethyl β-d-thiogalactopyranoside (PETG) (Invitrogen, Waltham, MA, USA) and 7-AAD. Afterwards, cells were filtered and analyzed on LSRII-CFI, VBGR 15-colour analyzer (Becton Dickinson, Mississauga, ON, Canada). Flow cytometry data were analyzed using flowjo (FlowJo, Ashland OR, USA).

### 4.11. RNA Isolation and Quantitative RT-PCR

Total RNA from IB3-1 cells transduced with HD-Ad-K18CFTR-TALEN or the control vector at 100 MOI were harvested and purified with the PureLink RNA Mini Kit (Life Technologies, Waltham, MA, USA) following the manufacturer’s instructions at 3 days, 6 passages, 12 passages, and 18 passages post-transduction, followed by DNase I digestion at room temperature for 15 min. One microgram of total RNA was reverse-transcribed using SuperScript VILO Master Mix (Invitrogen, Waltham, MA, USA) according to the manufacturer’s protocol. Ten nanograms of cDNA samples were loaded as templates for real-time PCR using Power SYBR Green PCR Master Mix on a QuantStudio 3 Thermal Cycler (Applied Biosystems, Foster City, CA, USA). The primers for detecting human CFTR cDNA were 5′ CCT GAG TCC TGT CCT TTC TC 3′ (forward primer) and 5′ CGC TGT CTG TAT CCT TTC CTC 3′ (reverse primer). GAPDH cDNA were measured as the internal control using primers 5′ GTT CGA CAG ACA GCC GTG TG 3′ (forward primer) and 5′ ATG GCG ACA ATG TCC ACT TTG C 3′ (reverse primer). Primers for detecting TALEN mRNA were 5′ GGG AGT GGA ACG AGT GAT TAG (forward primer) and 5′ ACT GAC TCA ACA GGT CAT CTT C (reverse primer). Relative levels of TALEN and hCFTR mRNA expression were calculated based the method of 2^−ΔΔ*C*T^.

### 4.12. Quantitation of Vector Genome

Total genomic DNA. Genomic DNA was extracted using the DNeasy blood and tissue kit (cat. 69504; Qiagen, Toronto, ON, Canada). Viral vector genome was quantified with qPCR using primers binding the viral ITR region 5′ CGG TGT ACA CAG GAA GTG ACA A (forward primer) and 5′ GCG GCC CTA GAC AAA TAT TAC G (reverse primer). Primers against the endogenous GAPDH gene were utilized as an internal control. Relative vector genome copy numbers were calculated based on the method of 2^−ΔΔ*C*T^.

### 4.13. CFTR Immunofluorescence

IB3-1 cells were seeded onto collagen-coated round glass coverslips in six-well plates 3 days prior to staining. At confluency, cells were washed in PBS, pH 7.4, fixed in ice-cold methanol for 10 min, and permeabilized with 0.5% Triton-100 in PBS. Cells were then blocked for 1 h in block solution (5% goat serum, 0.5% BSA, 0.05% Triton-100 in PBS). Cells were incubated with mouse monoclonal antibodies against the human *CFTR* R-domain MAB1660 and C-terminus MAB25031 (R&D Systems, Minneapolis, MN, USA) at 1:500 dilution overnight at 4 °C, washed in PBS with 0.05% Triton-100, and then incubated in the dark for 1 h in 1:750 diluted CF555 goat anti-mouse IgG (H+L) (Biotium, Fremont, CA, USA). After washing in the dark, cells on the glass slips were taken out from the six-well plates and mounted with one drop of VECTASHIELD HardSet Antifade Mounting Medium with DAPI (Vector Laboratories, Burlington, ON, Canada).

### 4.14. Protein Isolation and CFTR Immunoblot

IB3-1 cells were gently washed with ice-cold PBS, pH 7.4, then scraped from the bottom of the plate with a cell scraper in 1 mL of ice-cold PBS and centrifuged at 4°C. The cell pellet was resuspended on ice in 30 μL of RIPA buffer (1% Triton X-100, 0.1% SDS, 150 mMNaCl, 20 mM Tris-HCl, and 0.5% sodium deoxycholate) supplemented with cOmplete Protease Inhibitor Cocktail (Roche, Mississauga, ON, Canada). The cells were lysed on ice with occasional vortex, then centrifuged at 4 °C. Protein samples were denatured with one volume of 4× Laemmli protein sample buffer for SDS-PAGE (Bio-Rad, Mississauga, ON, Canada).

Protein samples were loaded onto polyacrylamide gel at 100 μL per lane and run at 100 volts for 90 min. The proteins in the gel were transferred onto a piece of nitrocellulose membrane at 85 V for 90 min on ice. The membrane was blocked in 5% skim milk in TBS-T (0.1% Tween 20 in TBS, pH 7.4) at room temperature with shaking, then incubated in 1:1000 diluted primary mouse anti-*CFTR* antibody MAB 596 (University of North Carolina) or 1:2000 diluted primary rabbit anti-GAPDH antibody ab9485 (Abcam, San Francisco, CA, USA) overnight at 4 °C. On the next day, membranes were washed with TBS-T and incubated with 1:3000 diluted secondary antibodies (Bio-Rad). The membranes were treated with Western Lightning Plus-ECL Solutions (PerkinElmer, Woodbridge, ON) and protein bands were visualized with ChemiDoc XRS+ (Bio-Rad, Mississauga, ON, Canada).

### 4.15. Protein Analysis with Automated Western System-JESS

Total protein isolation was carried out as described in the previous section. Protein samples (1 mg) were mixed with 5 × fluorescence detection master mix before loading onto each of the capillary reaction chamber. For CFTR protein detection, five microliters of 1:400 diluted anti-mouse CFTR antibody (MAB 596, University of North Carolina, Chapel Hill, NC, USA) and 5 μL of 1:400 diluted anti-rabbit GAPDH antibody (ab9485, Abcam) were used for each lane. For TALEN protein detection, 5 microliters of 1:400 diluted mouse-anti-flag antibody and 5 μL of 1:400 diluted anti-rabbit GAPDH antibody (ab9485, Abcam) were used for each lane. Ten microliters of combined secondary anti-M and anti-R rabbit antibody were used for each sample. The whole analysis including protein separation, primary and secondary antibody probing, and chemifluorescence detection, was done automatically using the JESS system (ProteinSimple, San Jose, CA, USA) under 5 h. Results from 24 samples were displayed and analyzed using the Compass analysis software.

### 4.16. FLIPR Assay

Protocol for FLIPR assay was adapted from Ahmadi et al. [45]. Cells were seeded to black-walled, clear-bottom 96-well plates and cultured until 100% confluency. The blue membrane potential dye (Molecular Devices) was dissolved in a chloride-free buffer (150 mM NMDG-gluconate, 3 mMKCl, 10 mM HEPES, pH 7.35, osmolarity 300 mOsm) and loaded to the cells. The plates were incubated at 37 °C with 5% CO_2_ and humidified air for 30 min, then transferred to the i3 multi-well microplate reader (Molecular Devices, San Jose, CA, USA). Eleven baseline reads were made, followed by addition of 2.5 μL forskolin (Sigma, Oakville, ON, Canada) per well. After adding the drug, 31 reads were made. Then, 21 scans were made after the reaction was terminated by addition of 10 μM CFTRinh-172. For negative controls, 2.5 microliters of dimethyl sulfoxide (DMSO; Sigma, Oakville, ON, Canada) was added instead of forskolin.

### 4.17. Iodide Efflux Assay

Transduced IB3-1 cells were seeded in 6 well plates and incubated for 1 h in iodide loading buffer (135 mM NaI, 4 mM KNO_3_, 2 mM Ca(NO_3_)·4H_2_O, 2 mM Mg(NO_3_)·6H_2_O, 11 mM Glucose, 20 mM Hepes). Iodide efflux was started by adding forskolin (40 µM) and measured at 1 min intervals using an iodide sensitive electrode (Thermo Fisher, Waltham, MA, USA).

### 4.18. Statistical Analysis

Unpaired Student’s *t*-test was used for the comparison between means of two groups. A *p*-value less than 0.05 was considered statistically significant. Error bars are shown as standard error of the mean (SEM).

## Figures and Tables

**Figure 1 genes-10-00039-f001:**
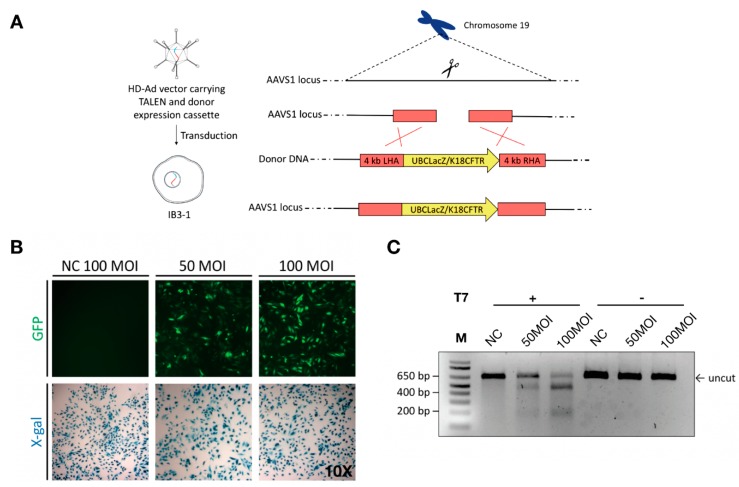
Co-delivery of TALENs and donor DNA using helper-dependent adenoviral (HD-Ad) vectors. (**A**) Schematic representation of HD-Ad vector-mediated integration of a gene cassette for expressing the LacZ reporter or human cystic fibrosis transmembrane conductance regulator (*CFTR*) into Exon 2 of the adeno-associated virus integration site 1 (AAVS1) locus. (**B**) Transduction of IB3-1 cells with a negative control (HD-Ad-K18LacZ) vector at 100 MOI and HD-Ad-UBCLacZ-TALEN vector at 50 and 100 MOI. X-gal staining detects cells with LacZ expression. (**C**) T7E1 assay for assessing the cleavage efficiency of cells transduced with the HD-Ad-K18LacZ-TALEN vector. Human IB3-1 cells were transduced with the vector at 50 MOI (Lane 2, 5) and 100 MOI (Lane 3, 6). Cells transduced with 100 MOI of HD-Ad-K18LacZ (Lane 1, 4) were used as negative controls. Lanes 1–3 were samples with T7 endonuclease treatment, while Lanes 4–6 were samples with no T7 endonuclease treatment. The arrow indicates the uncut PCR product.

**Figure 2 genes-10-00039-f002:**
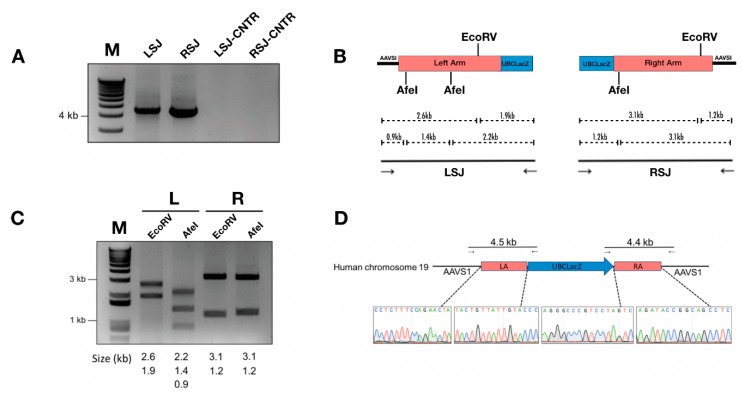
Characterization of site-specific gene integration. (**A**) Correct integration indicated by positive junction PCR product of 4.5 kb (left side junction, LSJ) and 4.4 kb (right side junction, RSJ). LSJ-CNTR and RSJ-CNTR are the left- and right-side junction PCR assessments of IB3-1 cells transduced with a 100 MOI of HD-Ad-UBCLacZ control vector. (**B**) Sanger sequencing of junction PCR products. (**C**) Restriction enzyme digestion of junction PCR products. Lane 1, left-side junction PCR product digested with EcoRV; Lane 2, left-side junction PCR product digested with AfeI; Lane 3, right-side junction PCR product digested with EcoRV; Lane 4, right-side junction PCR product digested with AfeI. (**D**) Restriction map showing the left-side junction and right-side junction PCR.

**Figure 3 genes-10-00039-f003:**
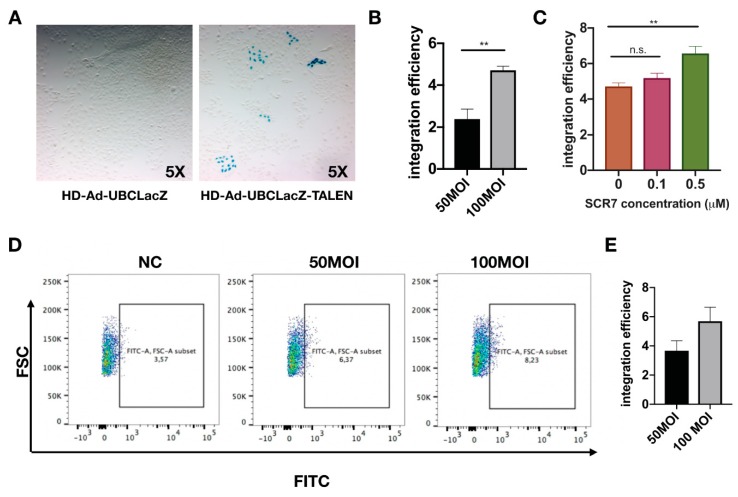
Analysis of the integration efficiency with the HD-Ad-UBCLacZ-TALEN vector. (**A**) X-gal staining for IB3-1 cells transduced with 100 MOI of HD-Ad-UBCLacZ-TALEN or HD-Ad-UBCLacZ at passage 18. (**B**) Quantification of integration efficiency of cells transduced with 50 MOI or 100 MOI of HD-Ad-UBCLacZ-TALEN; N = 3; ** *p* < 0.01. (**C**) Integration efficiency of HD-Ad-UBCLacZ-TALEN (100 MOI) in cells treated with different concentrations of SCR7. ** *p* < 0.01; n.s., *p* > 0.5. (**D**) Flow cytometry quantification of LacZ positive cells. IB3-1 cells transduced with 50 MOI or 100 MOI of HD-Ad-UBCLacZ-TALEN vector or 100 MOI of UBCLacZ vector were passaged for 18 generations and stained with Imagene green LacZ staining dye; NC, IB3-1 cells transduced with HD-Ad-UBCLacZ vector at 100 MOI. Percentages of LacZ positive cells were shown on the FITC vs. FSC plot, N = 3. (**E**) Bar graph showing integration efficiency subtracted from background (NC from (D)).

**Figure 4 genes-10-00039-f004:**
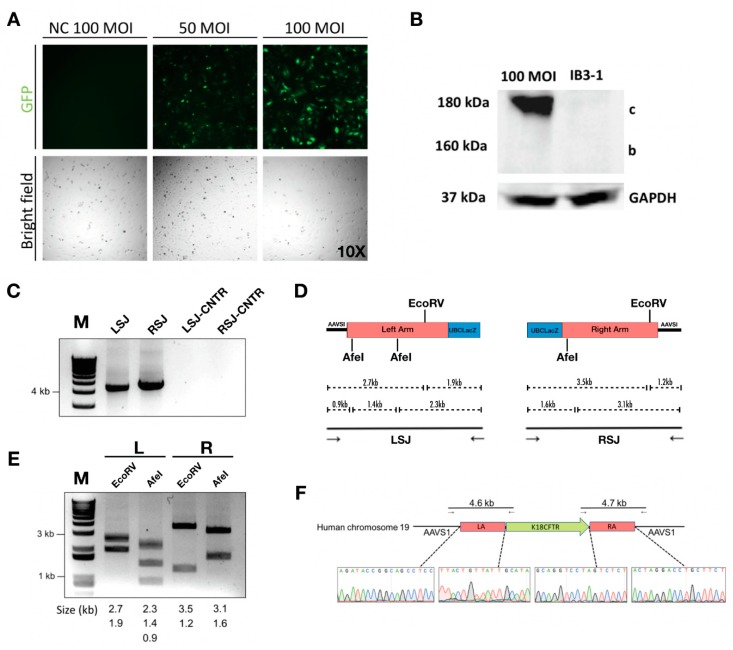
Integration of the human *CFTR* gene expression cassette into the AAVS1 locus. (**A**) Fluorescent images of IB3-1 cells transduced with 50 MOI or 100 MOI of HD-Ad-K18CFTR-TALEN vector; NC, untransduced cells. (**B**) Western blot for CFTR protein from cell lysate isolated 3 days post-transduction of 100 MOI of the HD-Ad-K18CFTR-TALEN vector. (**C**) Junction PCR analysis for IB3-1 cells transduced with the HD-Ad-K18CFTR-TALEN vector. Lane 1 shows the 4.6 kb product from the left-side junction and lane 2 shows the 4.7 kb product from the right-side junction. Lanes 3 and 4 are negative control PCR reactions for the left- and right-side junctions, respectively. (**D**) Restriction analysis of the junctional PCR products. Lane 1 and 2 are the left-side junction PCR product digested with EcoRV and AfeI, respectively; Lane 3 and 4 are the right-side junction PCR product digested with EcoRV and AfeI, respectively. (**E**) Restriction map for left-side and right-side junction PCR. (**F**) Right panel, Sanger sequencing around junction site of the PCR products.

**Figure 5 genes-10-00039-f005:**
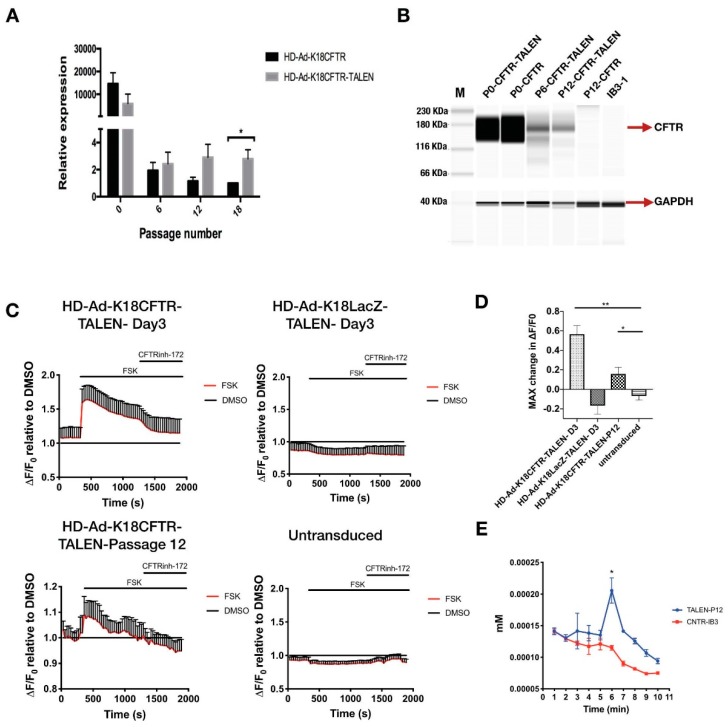
Analysis of *CFTR* mRNA and protein expression, and channel function. (**A**) qPCR quantification of CFTR mRNA expression from IB3-1 cells transduced with 100 MOI of HD-Ad-K18CFTR-TALEN vector compared to the control vector (without TALEN) at different passages. * *p* < 0.05. (**B**) CFTR western performed using the JESS system. IB3-1 cells were transduced with 100 MOI of HD-Ad-K18CFTR-TALEN vector or 100 MOI of HD-Ad-K18CFTR vector. Lysates were collected at passage 0, passage 6, and passage 12. (**C**) CFTR channel function was measured using a FLIPR assay of IB3-1 cells transduced with 100 MOI of HD-Ad-K18-CFTR-TALEN vector, HD-Ad-K18LacZ TALEN vector, and untransduced cells. (**D**) Bar graph representing maximum change in ΔF/F_0_ after addition of agonist relative to baseline measurement. N = 3, * *p* < 0.05, ** *p* < 0.01. (**E**) Iodide efflux assay for IB3-1 cells transduced with HD-Ad-K18CFTR-TALEN vector and passed for 12 generations. CNTR-IB3-1, cells without transduction. N = 3, * *p* < 0.05.

**Figure 6 genes-10-00039-f006:**
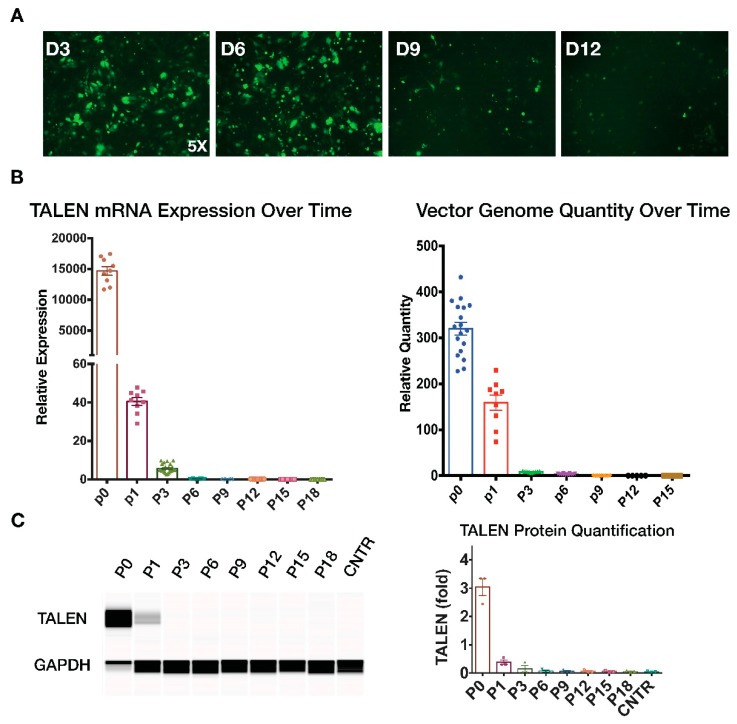
Disappearance of TALEN expression and vector genome over time. (**A**) GFP fluorescence at different time points following HD-Ad-K18CFTR-TALEN vector transduction 100 MOI. (**B**) qPCR quantification of TALEN mRNA levels, as well as vector genome in IB3-1 cells transduced with HD-Ad-K18CFTR-TALEN vector. N = 3. (**C**) Western blot analysis of TALEN protein expression at different time points post transduction using the JESS system.

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
