# Peer review of "TALEN-Mediated Gene Targeting for Cystic Fibrosis-Gene Therapy"

_genes, 2019, doi:10.3390/genes10010039_

Round 1

Reviewer 1 Report

“TALEN mediated gene targeting for CF gene therapy” by Xia et al reports the generation of HD-AD vectors to deliver gene editing tools to IB3 cells targeting the AAVS1 locus. The authors demonstrate integration of both LacZ and CFTR by sequencing and serial passage. Efficiencies ranged from 3-8%.

Comments

Figure 2A, middle panel:  It’s difficult to interpret these data without a perspective of where EcoRV and Afe1 are located in the genome and the homology arms.

Supplemental Figure 1:  The screenshots of the plasmid maps are difficult to read – although I appreciate that the elements are to scale. A well-labeled linear map with the relevant restriction enzyme sites and sizes of the gene elements would be easier to interpret.

Figure 3C: What was the MOI?

Figure 5C: fourth panel:  For clarity, “TALEN-P12” should be relabeled “HD-Ad-K18CFTR-TALEN-Passage 12”.

Figure 6: There are no letters for the panels. The methods section doesn’t include details for these experiments.  Currently it is unclear what the TALEN mRNA or Vector genomes are being compared to.

Supplemental Figure 5: These data could be deleted.

Line 96: This statement should read: “… no detectable off-target activity in the 3 bioinformatically predicted genomic loci.”

Line 116-117: These lines should be clarified. Figure 4A is GFP and Figure 4B is CFTR.  Currently, the text suggests that both figures indicate CFTR protein.

Author Response

We thank the reviewers for their constructive comments and provided our specific responses below:

Reviewer 1

“TALEN mediated gene targeting for CF gene therapy” by Xia et al reports the generation of HD-AD vectors to deliver gene editing tools to IB3 cells targeting the AAVS1 locus. The authors demonstrate integration of both LacZ and CFTR by sequencing and serial passage. Efficiencies ranged from 3-8%. 

Comments

Figure 2A, middle panel:  It’s difficult to interpret these data without a perspective of where EcoRV and Afe1 are located in the genome and the homology arms.

Response: Schematic diagrams of the restriction enzyme cutting sites within the genome and homology arms have now been provided.

Supplemental Figure 1:  The screenshots of the plasmid maps are difficult to read – although I appreciate that the elements are to scale. A well-labeled linear map with the relevant restriction enzyme sites and sizes of the gene elements would be easier to interpret.

Response: Linearized versions of plasmid maps have now been provided.

Figure 3C: What was the MOI?

Response: We have added the MOI (100) to the legend of Figure 3C.

Figure 5C: For clarity, “TALEN-P12” should be relabeled “HD-Ad-K18CFTR-TALEN-Passage 12”.

Response: The change was made in Figure 5C.

Figure 6: There are no letters for the panels. The methods section doesn’t include details for these experiments.  Currently it is unclear what the TALEN mRNA or Vector genomes are being compared to.

Response: Panel labels in Figure 6 have now been provided. The methods section was modified with added details related to Figure 6. The TALEN mRNA and vector genome values were compared to that of cells at Passage 0 following transduction.

Supplemental Figure 5: These data could be deleted.

Response: These data were included to provide additional information supporting Figure 6.

Line 96: This statement should read: “… no detectable off-target activity in the 3 bioinformatically predicted genomic loci.”

Response: Line 96 was modified accordingly.

Line 116-117: These lines should be clarified. Figure 4A is GFP and Figure 4B is CFTR.  Currently, the text suggests that both figures indicate CFTR protein.

Response: Figure 4A and 4B were describing the CFTR vector where the GFP gene was carried in the vector. Only in Figure 4B we examine CFTR protein expression; we made changes to the text.

Reviewer 2 Report

This is a very interesting work, with an attractive technique of gene editing for the therapy of CF patients. However, sometimes the manuscript lacks in clarity and would benefit of a text revision to better explain the results and their relevance.

Some minor comments follow:

- Since not all the readers may have expertise in this field, it is better to expain the meaning of AAVSI and MOI, or of other abbreviations used in the text.

- More information about the state of the art of the different therapeuthic approaches available for cystic fibrosis, including gene editing, gene therapy and drug tratment approaches, should be added in the introduction.

- Please provide more information about the CFTR genotype and the origin of the cell lines used.

- Please explain the choice of FLIPR assay and iodide efflux measure to test for CFTR activity

- Methodological information, or at least the restriction maps, are required to understand restriction analyses presented in figures 2A and 4C.

- Figure 6: letters labeling the different panels are missing.

Author Response

Reviewer 2 (comments came last)

This is a very interesting work, with an attractive technique of gene editing for the therapy of CF patients. However, sometimes the manuscript lacks in clarity and would benefit of a text revision to better explain the results and their relevance.

Some minor comments follow:

- Since not all the readers may have expertise in this field, it is better to expain the meaning of AAVSI and MOI, or of other abbreviations used in the text.

Response: We thank you for the comments. Another reviewer also raised this concern, and we have made the changes accordingly.

- More information about the state of the art of the different therapeuthic approaches available for cystic fibrosis, including gene editing, gene therapy and drug tratment approaches, should be added in the introduction.

Response: This is a very good suggestion. Considering this research article is submitted to a special issue on Cystic Fibrosis, where other review articles will discuss all different techniques and therapies, we will keep our introduction concise to avoid repetition.

- Please provide more information about the CFTR genotype and the origin of the cell lines used.

Response: This has also been raised by another reviewer. We have made the changes in the methods section accordingly.

- Please explain the choice of FLIPR assay and iodide efflux measure to test for CFTR activity

Response: We explained the flipr and iodide efflux assays in methods.

- Methodological information, or at least the restriction maps, are required to understand restriction analyses presented in figures 2A and 4C.

Response: This has also been pointed out by another reviewer and we have made the changes.

- Figure 6: letters labeling the different panels are missing.

Response: We added the letters to Figure 6.

Reviewer 3 Report

The subject addressed in the manuscript is well framed in the recent renewal interest for cystic fibrosis gene therapy, which is based on the concept that gene complementation/replacement would be effective in correcting CF defects irrespective of gene mutations underlaying the disease.

The authors, driven by the hypothesis that adenoviral-based vector may efficiently deliver an integrating-CFTR expression cassette in epithelial cells, constructed TALEN-based integrative Ad-vectors with reporter or CFTR cDNAs. The efficiency of the vectors was tested in CF epithelial cells IB3. According with the data herein reported, it appears that TALEN-directed integration of a CFTR minigene may provide long term gene expression in replicating cells, as assessed by the analysis of CFTR gene expression (both mRNA and protein levels) and the functional assays. Indeed, the results show a stable increase, of about 3-fold, of CFTR mRNA from passage 6 to passage 12 giving rise to similar amount of mature CFTR protein, but very low functional activity at passage 12. Whether this activity would be sufficient to complement CFTR dysfunction is hard to predict as no positive control was included. Therefore, the major weakness in this manuscript is the complete absence of a positive control expressing functional CFTR, which is absolutely required to provide any predictive functional activity of the integrated CFTR transgene. An example of CFTR expression and functional activity in IB3 cells is reported in Ye et al, 2001. (Ye L, et al. Regulated expression of the human CFTR gene in epithelial cells. Mol Ther. 2001 May;3(5 Pt 1):723-33).

Minor points

Abbreviation should be specified, i.e AAVSI

Introduction should be improved by considering the following recent references on the revision of the corrector/potentiator treatment of CF patients and new approaches for CF gene therapy.

Southern KW, et al., Correctors (specific therapies for class II CFTR mutations) for cystic fibrosis. Cochrane Database Syst Rev. 2018 Aug 2;8:CD010966. doi: 10.1002/14651858.CD010966.pub2

Haque AKMA et al, Chemically modified hCFTR mRNAs recuperate lung function in a mouse model of cystic fibrosis. Sci Rep. 2018 Nov 13;8(1):16776. doi: 10.1038/s41598-018-34960-0.

Cooney AL et al. Widespread airway distribution and short-term phenotypic correction of cystic fibrosis pigs following aerosol delivery of piggyBac/adenovirus. Nucleic Acids Res. 2018 Oct 12;46(18):9591-9600. doi: 10.1093/nar/gky773.

De Rocco D, et al Assembly and Functional Analysis of an S/MAR Based Episome with the Cystic Fibrosis Transmembrane Conductance Regulator Gene. Int J Mol Sci. 2018 Apr 17;19(4). pii: E1220. doi: 10.3390/ijms19041220.

Results.

Pag 2, line 88 - The rationale underlaying the T7E1 assay should be added and/or appropriate reference.

Pag3, line 97 – “to determine the integration efficiency of LacZ vector ….. cells were passage for 18 generations”. This experimental plan is a typical long term kinetics to study stability, rather than integration efficiency.  Unless the authors hypothesize cycle of integration-excision, not supported by the TALEN expression study (Fig 5) it is expected that once the cassette is integrated it will be stably maintained. Of course, this does not guarantee CFTR expression which may be silenced with time.

Material and methods

The CFTR allele of IB3 cells should be added, as well as the provider. Typically, IB3 cells are grown with LHC-8, did the authors analyzed cells in DMEM as respect to LHC-8?

Pag 7 line 315, relative quantification by the 2-DDCT require a calibrator, that is not indicated.

Statistical analyses of the functional assays, membrane potential assay and iodide-efflux is missing.

Author Response

Reviewer 3

The subject addressed in the manuscript is well framed in the recent renewal interest for cystic fibrosis gene therapy, which is based on the concept that gene complementation/replacement would be effective in correcting CF defects irrespective of gene mutations underlaying the disease.

The authors, driven by the hypothesis that adenoviral-based vector may efficiently deliver an integrating-CFTR expression cassette in epithelial cells, constructed TALEN-based integrative Ad-vectors with reporter or CFTR cDNAs. The efficiency of the vectors was tested in CF epithelial cells IB3. According with the data herein reported, it appears that TALEN-directed integration of a CFTR minigene may provide long term gene expression in replicating cells, as assessed by the analysis of CFTR gene expression (both mRNA and protein levels) and the functional assays. Indeed, the results show a stable increase, of about 3-fold, of CFTR mRNA from passage 6 to passage 12 giving rise to similar amount of mature CFTR protein, but very low functional activity at passage 12. Whether this activity would be sufficient to complement CFTR dysfunction is hard to predict as no positive control was included. Therefore, the major weakness in this manuscript is the complete absence of a positive control expressing functional CFTR, which is absolutely required to provide any predictive functional activity of the integrated CFTR transgene. An example of CFTR expression and functional activity in IB3 cells is reported in Ye et al, 2001. (Ye L, et al. Regulated expression of the human CFTR gene in epithelial cells. Mol Ther. 2001 May;3(5 Pt 1):723-33).

Response: This is a fair comment. A correct positive control would allow the assessment of the degree of restoration of CFTR activity in the transduced cells. The Tet-inducible CFTR expression cell lines we created (Ye L, 2001 Mol Ther, 3:723-33) based on IB3-1 cells express too much CFTR. The ideal control is to select a single CFTR-positive cell clone from our transduced cells. This requires much more time to complete the experiment. We will consider it in our future studies.

Minor points

Abbreviation should be specified, i.e AAVSI

Response: AAVSI locus has now been specified in the introduction.

Introduction should be improved by considering the following recent references on the revision of the corrector/potentiator treatment of CF patients and new approaches for CF gene therapy.

Southern KW, et al., Correctors (specific therapies for class II CFTR mutations) for cystic fibrosis. Cochrane Database Syst Rev. 2018 Aug 2;8:CD010966. doi: 10.1002/14651858.CD010966.pub2

Response: This has been cited as Ref #10.

Haque AKMA et al, Chemically modified hCFTR mRNAs recuperate lung function in a mouse model of cystic fibrosis. Sci Rep. 2018 Nov 13;8(1):16776. doi: 10.1038/s41598-018-34960-0.

Response: This has been cited as Ref #22.

Cooney AL et al. Widespread airway distribution and short-term phenotypic correction of cystic fibrosis pigs following aerosol delivery of piggyBac/adenovirus. Nucleic Acids Res. 2018 Oct 12;46(18):9591-9600. doi: 10.1093/nar/gky773.

Response: This has been cited as Ref #16.

De Rocco D, et al Assembly and Functional Analysis of an S/MAR Based Episome with the Cystic Fibrosis Transmembrane Conductance Regulator Gene. Int J Mol Sci. 2018 Apr 17;19(4). pii: E1220. doi: 10.3390/ijms19041220.

Response: This has been cited as Ref #23.

Results.

Pag 2, line 88 - The rationale underlaying the T7E1 assay should be added and/or appropriate reference.

Response: The T7E1 assay was used to determine TALEN cleavage efficiency and a reference has been added in the main text.

Pag3, line 97 – “to determine the integration efficiency of LacZ vector ….. cells were passage for 18 generations”. This experimental plan is a typical long term kinetics to study stability, rather than integration efficiency.  Unless the authors hypothesize cycle of integration-excision, not supported by the TALEN expression study (Fig 5) it is expected that once the cassette is integrated it will be stably maintained. Of course, this does not guarantee CFTR expression which may be silenced with time.

Response: We agree that the integrated gene should be stable. Because we do not expect gene integration to occur in all transduced cells, passaging was implemented to dilute out the residual vector in cells. At P18, all detectable LacZ expression was expected to be from integrated LacZ gene.  

Material and methods

The CFTR allele of IB3 cells should be added, as well as the provider. Typically, IB3 cells are grown with LHC-8, did the authors analyzed cells in DMEM as respect to LHC-8?

Response: We did use LHC-8 for IB3-1 cells when we created new cell lines (Ye et al 2001, Mol Ther 3-723-733). However, for gene expression studies, we cultured IB3-1 cells in DMEM because we did not see advantages of using LHC-8.

Pag 7 line 315, relative quantification by the 2-DDCT require a calibrator, that is not indicated.

Response: 2-DDCT was calibrated based on the Ct value of the negative control, and this information has been added.

Statistical analyses of the functional assays, membrane potential assay and iodide-efflux is missing.

Response:  Statistical analyses have been added.

Reviewer 4 Report

This is a very interesting paper with an intriguing technique for gene therapy in patients with cystic fibrosis. 

However, the data at times lacks clarity, and therefore I have some minor comments.

1.    The CFTR minigene is referenced but not described. Is this simply a humanized cDNA? Or is it truncated in some form? Please provide more detail either in the introduction or the methods section.

2.    There are minor typos and grammatical errors throughout.  Please look over in detail.

3.    In the results sections, Figure 1C, 2A, and 4C, please label the bands on the figure. While this information is in the legend, it is still hard to distinguish and labeling on the image will greatly ease the readers understanding.

4.    It says in the methods that cells were transduced at P0, but nowhere in the body of the results. Since this is quite important, please add to the relevant results sections.

5.    Are the membrane potential assays the same as the FLIPR assays? This is not clear. If not, please state what the FLIPR assays are measuring.

6.    The FLIPR assay is not a traditional measure of CFTR activity. Therefore, I think it would be helpful to have the iodide-efflux data moved from the supplement to Figure 5. 

7.    While Figure 5 clearly shows a response to forskolin over untransduced and LacZ, it is hard to understand the meaningfulness of this increase without WT data. Please include data conducted on these assays in a WT cell line.

8.    Please give more detail about the cell line in the methods, beyond just the citation. Were these collected from a F508del patient? Is there a WT control for this line? Etc. 

9.    Also, please add methods for the iodide-efflux assay.

Author Response

Reviewer 4

This is a very interesting paper with an intriguing technique for gene therapy in patients with cystic fibrosis. 

However, the data at times lacks clarity, and therefore I have some minor comments.

1.    The CFTR minigene is referenced but not described. Is this simply a humanized cDNA? Or is it truncated in some form? Please provide more detail either in the introduction or the methods section.

Response: The CFTR minigene is the normal CFTR cDNA and its expression is driven by the human cytokeratin 18 gene promoter as described in the reference. This information has now been added to the abstract and the results section.

2.    There are minor typos and grammatical errors throughout.  Please look over in detail.

Response: Minor typos and grammatical errors were corrected by spell check and proof-reading.

3.    In the results sections, Figure 1C, 2A, and 4C, please label the bands on the figure. While this information is in the legend, it is still hard to distinguish and labeling on the image will greatly ease the readers understanding.

Response: The bands in Figure 1C, 2A, and 4C have now been relabeled.

4.    It says in the methods that cells were transduced at P0, but nowhere in the body of the results. Since this is quite important, please add to the relevant results sections.

Response: Description of transduction was added to the results section.

5.    Are the membrane potential assays the same as the FLIPR assays? This is not clear. If not, please state what the FLIPR assays are measuring.

Response: FLIPR and membrane potential assays were describing the same thing and this information was clarified in the main text.

6.    The FLIPR assay is not a traditional measure of CFTR activity. Therefore, I think it would be helpful to have the iodide-efflux data moved from the supplement to Figure 5. 

Response: Iodide efflux assay was moved from the supplement to Figure 5.

7.    While Figure 5 clearly shows a response to forskolin over untransduced and LacZ, it is hard to understand the meaningfulness of this increase without WT data. Please include data conducted on these assays in a WT cell line.

Response: Levels of CFTR activity in different normal individuals are different. The ideal wild type control is to correct the CFTR in the mutant cells used for experiments. Considering the time needed for generating the line, we could not finish the experiments within the allowed time for manuscript revision. We will consider this in our future studies. We think that our current level of gene correction is low and have discussed how to enhance the efficiency in the discussion section. 

8.    Please give more detail about the cell line in the methods, beyond just the citation. Were these collected from a F508del patient? Is there a WT control for this line? Etc. 

Response: Cell line information was added to the methods section.

9.    Also, please add methods for the iodide-efflux assay.

Response: The method for iodide-efflux assay was added.

Reviewer 5 Report

This paper describes the development of gene editing by gene therapy for cystic fibrosis with TALENs delivered by HDAd. The authors have a lot of experience in the field of CF and HDAd in particular. Despite the development of small molecule drugs for CF, a significant proportion of patients have non-druggable mutations and so require other therapies. Also, gene editing is still in early stages of development and new delivery technologies in particular are of interest. Therefore, I believe this paper will be of wide interest although I have significant reservations about the manuscript and some of the experiments described in it/

Introduction

There have been several published reports of gene editing approaches for cystic fibrosis including by use of TALENS so these must be reported and referenced to provide the correct context for this study. 

Figure 1

GFP data is presented in the Fig 1B but this is not discussed in the main text. Please correct this. Also, GFP is described in the legend as a conjugate  with  TALEN. This needs further explanation. Is this a fusion in which case a map diagram is needed and some evidence that TALEN in a fusion is still functional; or a biochemical conjugate in which case the methodology is required.

Figure 2, middle. It is not possible to evaluate the restriction digests without a restriction map for EcoRV and Afe I so this should be provided in the map on the right.

Fig. 2B - T7E1 assay is not sufficiently sensitive to assess off-target effects especially in a mixed population of treated cells so this result is pretty meaningless. This figure should be removed and should be replaced by PCR, Sanger sequencing and TIDE or ICR analysis as a minimum standard.

Fig. 3  The NC values should be subtracted from the substrate assay and FACS values to yield the correct integration value. 

SCR7 is now rarely used as it was recognised that it is not specific for DNA Ligase and has other effects so this should be recognised in the text. 

What Moi was used in Fig. 3C? Please correct legend.

Fig 4 As in Fig 1, the GFP data needs a lot more explanation. 

4C. A restriction map showing EcoRV and AfeI and the PCR primers is required to make sense of the restriction digest data. An estimation of the CFTR insertion rate should be made. 

Fig 5 – The increase in the level of CFTR mRNA is only significant at passage 18 at 3-fold. if the efficiency of integration is similar to lacZ at 5%, is that sufficient to correct the CF defect? Other reports suggest more like 10% in over-expression systems. To better assess functional correction, cell clones should have been screened and tested rather than the mixed population where corrections rates are very low. Thsis data should be explained and discussed for relevance in the context of the low level of correction or data supporting higher levels fo correction shown

Fig 6. Again, further explanation of the GFP data is required. Passage data is not really relevant to in vivo correlation, which is the subject of the discussion fo the data, but rather than cell divisions so please present the data with estimated cell divisions. E.g., for a 1/7 dilution at passage, about 3 passages will occur per passage. Thus, the risks of off atrgete events and immunogenicity may be more significant than originally estimated.

Overall this paper is of interest, particularly for the delivery methods but the functional data in the context of the low level of correction needs further justification, the evaluation of OTE requires more sensitiive methods and the GFP data requires further details.

Author Response

Reviewer 5

This paper describes the development of gene editing by gene therapy for cystic fibrosis with TALENs delivered by HDAd. The authors have a lot of experience in the field of CF and HDAd in particular. Despite the development of small molecule drugs for CF, a significant proportion of patients have non-druggable mutations and so require other therapies. Also, gene editing is still in early stages of development and new delivery technologies in particular are of interest. Therefore, I believe this paper will be of wide interest although I have significant reservations about the manuscript and some of the experiments described in it/

Introduction

There have been several published reports of gene editing approaches for cystic fibrosis including by use of TALENS so these must be reported and referenced to provide the correct context for this study. 

Response: Two references on gene editing approaches for cystic fibrosis using TALEN were cited in the introduction.

Figure 1

GFP data is presented in the Fig 1B but this is not discussed in the main text. Please correct this. Also, GFP is described in the legend as a conjugate  with  TALEN. This needs further explanation. Is this a fusion in which case a map diagram is needed and some evidence that TALEN in a fusion is still functional; or a biochemical conjugate in which case the methodology is required.

Response: GFP data presented in Figure 1B has now been described in the results section. GFP was expressed from a separated (CMV) promoter and vector maps were included in supplementary Figure 1.

Figure 2, middle. It is not possible to evaluate the restriction digests without a restriction map for EcoRV and Afe I so this should be provided in the map on the right.

Response: A restriction map was provided for figure 2.

Fig. 2B - T7E1 assay is not sufficiently sensitive to assess off-target effects especially in a mixed population of treated cells so this result is pretty meaningless. This figure should be removed and should be replaced by PCR, Sanger sequencing and TIDE or ICR analysis as a minimum standard.

Response: We agree that T7E1 assay is not sensitive enough although it can detect off-target effects if an off-target site has a high frequency of cleavage. A non-biased, genome-wide analysis would be useful. We do not have enough time to get the experiments done.

Fig. 3  The NC values should be subtracted from the substrate assay and FACS values to yield the correct integration value. 

Response: The NC values were now subtracted from the FACS values.

SCR7 is now rarely used as it was recognised that it is not specific for DNA Ligase and has other effects so this should be recognised in the text. 

Response:  Thank you for pointing this out. We modified the text and cite a reference to recognize this in the main text.

What Moi was used in Fig. 3C? Please correct legend.

Response:  The MOI (100) was now added to the figure legend.

Fig 4 As in Fig 1, the GFP data needs a lot more explanation. 

Response: Description regarding GFP data has now been added to Figure 1 and Figure 4A.

4C. A restriction map showing EcoRV and AfeI and the PCR primers is required to make sense of the restriction digest data. An estimation of the CFTR insertion rate should be made. 

Response: Restriction maps were inserted in Figure 2. The CFTR insertion rate is expected to be similar to that of the LacZ gene since the homology arms are the same in size. The purpose of using the LacZ reporter gene is to assess the insertion rate.

Fig 5 – The increase in the level of CFTR mRNA is only significant at passage 18 at 3-fold. if the efficiency of integration is similar to lacZ at 5%, is that sufficient to correct the CF defect? Other reports suggest more like 10% in over-expression systems. To better assess functional correction, cell clones should have been screened and tested rather than the mixed population where corrections rates are very low. Thsis data should be explained and discussed for relevance in the context of the low level of correction or data supporting higher levels fo correction shown

Response: The gene integration efficiency (5%) achieved in this study is not enough for fully correcting the CF defect. This should be considered as the low level of correction. There is a need to increase the efficiency. We mentioned this in the discussion section.

Fig 6. Again, further explanation of the GFP data is required. Passage data is not really relevant to in vivo correlation, which is the subject of the discussion fo the data, but rather than cell divisions so please present the data with estimated cell divisions. E.g., for a 1/7 dilution at passage, about 3 passages will occur per passage. Thus, the risks of off atrgete events and immunogenicity may be more significant than originally estimated.

Response: We added the information on GFP data to the relevant results sections. We have also added the information of cell dilution at passages to the methods. Risks associated with off-target activity and immunogenicity in vivo were not the focus in this study and will be examined in the future.

Overall this paper is of interest, particularly for the delivery methods but the functional data in the context of the low level of correction needs further justification, the evaluation of OTE requires more sensitiive methods and the GFP data requires further details.

Response: We thank you for your constructive comments. We have responded to all of them individually.

Round 2

Reviewer 1 Report

The authors addressed my prior concerns.

Author Response

Thank you for spending your time to review our manuscript and your constructive comments. 

Reviewer 3 Report

Regarding the first answer there are other possible positive controls such as the already described corrected-IB3 or cells with different genetic background but expressing CFTR, such as 16HBE. 

Irrespective of this, the author answered all the questions and the manuscript is acceptable for publications

Author Response

(The authors gave the same response as above.)

Reviewer 4 Report

Comments have been addressed.

Author Response

(The authors gave the same response as above.)

Reviewer 5 Report

The authors have addressed most of the points from my original report but some issues have not been addressed. 

Fig. 2B T7 assay for off-target effects. It is good they think about this but i am still not satisfied with their response to my comments - T7 is simply not an acceptable assay for off-target effects and the data is pretty meaningless so I think this figure should be removed or the text should be modified to explain the limitations of the analysis to detecting only high -frequency off target events. 

Author Response

Thank you for spending your time to review our manuscript and your constructive comments. 

I think that in your comments, you meant Fig. 2E, not Fig. 2B, that needs to be deleted, since Fig.2E shows the T7E1 assay results.  We deleted that part and modified the main next and the figure legend accordingly.